# Antifungal Activities of Sulfur and Copper Nanoparticles against Cucumber Postharvest Diseases Caused by *Botrytis cinerea* and *Sclerotinia sclerotiorum*

**DOI:** 10.3390/jof8040412

**Published:** 2022-04-16

**Authors:** Mohamed E. Sadek, Yasser M. Shabana, Khaled Sayed-Ahmed, Ayman H. Abou Tabl

**Affiliations:** 1Plant Pathology Department, Faculty of Agriculture, Mansoura University, El-Mansoura 35511, Egypt; scare_mmm@yahoo.com (M.E.S.); ayman@mans.edu.eg (A.H.A.T.); 2Department of Agricultural Chemistry, Faculty of Agriculture, Damietta University, New Damietta 34517, Egypt

**Keywords:** S-NPs, Cu-NPs, antifungal activity, cytotoxicity, cucumber, *Botrytis cinerea*, *Sclerotinia sclerotiorum*

## Abstract

Nanoparticles (NPs) have attracted great interest in various fields owing to their antimicrobial activity; however, the use of NPs as fungicides on plants has not been sufficiently investigated. In this study, the antifungal activities of sulfur nanoparticles (S-NPs) and copper nanoparticles (Cu-NPs) prepared by a green method were evaluated against *Botrytis cinerea* and *Sclerotinia sclerotiorum*. The formation of NPs was confirmed by transmission electron microscopy (TEM) and X-ray diffraction analysis (XRD). The antifungal activities of NPs (5–100 µg/mL), CuSO_4_ (4000 µg/mL), and micro sulfur (MS) were compared to those of the recommended chemical fungicide Topsin-M 70 WP at a dose of 1000 µg/mL. They were evaluated in vitro and then in vivo at different temperatures (10 and 20 °C) on cucumber (*Cucumis sativus*) fruits. The total phenolic content (TPC) and total soluble solids (TSS) were determined to study the effects of various treatments on the shelf life of cucumber fruits, compared to untreated cucumber as a positive control. The diameters of S-NPs and Cu-NPs ranged from 10 to 50 nm, and 2 to 12 nm, respectively. The results revealed that S-NPs exhibited the highest antifungal activity, followed by Cu-NPs. However, CuSO_4_ showed the lowest antifungal activity among all treatments. The antifungal activity of the prepared NPs increased with the increase in NP concentration, while the fungal growth was less at low temperature. The cytotoxicity of the prepared NPs was evaluated against the WI-38 and Vero cell lines in order to assess their applicability and sustainability. S-NPs caused less cytotoxicity than Cu-NPs.

## 1. Introduction

In recent decades, nanotechnology has attracted great interest across a wide range of fields. The application of nanotechnology demands the integration of knowledge from chemistry, physics, biology, engineering, and material science [1,2]. Agriculture is one of the most important economic sectors for food security worldwide; within this sector, plant diseases pose a significant threat to plant production and sustainability. Nanotechnology may lessen the environmental risks caused by chemical pesticides and fertilizers, and it has been found that nano-pesticides/fertilizers are more effective at low doses [3,4,5,6]. The use of hazardous chemical pesticides has severe effects on soil microbes and animal health. The field of bionanotechnology has been able to introduce sufficient, eco-friendly, alternative antifungal agents in the agriculture sector [6].

*Botrytis cinerea* is a causal agent of gray mold disease that decreases the productivity of cucumbers and tomatoes grown in greenhouses by affecting the fruits, stems, and leaves [7]. *S**clerotinia*
*sclerotiorum* causes white mold, which is a common disease in cucumbers and beans grown in all production regions. Both diseases lead to a serious loss in yield of up to 100% when the optimum weather conditions for pathogens are met [8]. It has been reported that *S. sclerotiorum*, which infects about 600 plant species [9], can stay in the soil in the form of sclerotia for approximately 10 years [10]. The use of fungicides and conventional practices represent the most common strategies. Fungicides are widely used to minimize loss in crop production. However, over the last 15 years, agrochemical scrutiny has attracted great interest; therefore, numerous fungicides have been banned [11,12]. 

It is preferable to considerably decrease the amount of fungicides used so that cost is minimized and, at the same time, the resistance of pathogens to the fungicides can be reduced. NPs exhibit high specific area, so the use of NPs may minimize the required amounts of antifungal agents [3]. Bulk, micro, and nanoscale sulfur have a wide range of applications in various industrial facets, including sulfuric acid production, fertilizers, and antimicrobial agents [13]. Sulfur can be used in agriculture as a fungicide against many fungal diseases that affect grapes, apples, strawberries, vegetables, and other cultivated plants. It is considered a highly efficient fungicide in agriculture [14].

Copper, as a promising metal with antimicrobial activity, can also be used as a fungicide. Some studies have investigated the biogenic synthesis of Cu-NPs [15,16]. Copper and its salts have been widely utilized as potential antimicrobial agents [17] in agriculture as pesticides, fungicides, herbicides, and algaecides [18,19,20,21]. Nanotechnology improves the antimicrobial activity of sulfur and copper by manipulating them into nanoparticles [3]. 

This study aimed to utilize nanoparticles as eco-friendly, cost-effective, and adequate fungicides against *B. cinerea* and *S. sclerotiorum* on cucumbers. In this respect, the antifungal activity of green-synthesized S-NPs and Cu-NPs against these fungi was evaluated in vitro and in vivo in comparison with the common fungicide Topsin-M 70 WP and micro sulfur in addition to copper salt (CuSO_4_). The cytotoxicity of the prepared NPs was also evaluated to confirm their applicability as safe and sustainable fungicides for fungal disease control. 

## 2. Experimental 

### 2.1. Materials

Copper (II) sulfate pentahydrate (CuSO_4_.5H_2_O), polyvinylpyrrolidone (PVP), ascorbic acid, oxalic acid, sodium thiosulfate pentahydrate (Na_2_S_2_O_3_·5H_2_O), and potato dextrose agar (PDA) medium were purchased from Sigma-Aldrich and used without further purification. 

### 2.2. Synthesis of Nanoparticles

#### 2.2.1. Green Synthesis of Cu-NPs

Cu-NPs were synthesized via a green manner [22] with an improved modification. Cu-NPs were prepared using copper (II) sulfate as a precursor in the presence of PVP as a stabilizer. Ascorbic acid was used as an eco-friendly reductant to reduce CuSO_4_ into Cu-NPs. PVP was used as an eco-friendly stabilizer and dissolved in a CuSO_4_ solution (0.5 M) at a concentration of 6 g/100 mL, and then ascorbic acid solution (1 M) was added to this mixture at a volume ratio of 1:1 under continuous stirring for 4 h. The color of the solution changed from blue to yellow to brown, indicating the formation of Cu-NPs and the complete reduction of CuSO_4_. 

#### 2.2.2. Preparation of S-NPs 

S-NPs were prepared through a precipitation reaction based on the method described by Chaudhuri et al. [14] with an improved modification. PVP was dissolved in an oxalic acid solution (5 mM) at a concentration of 6 g PVP/100 mL. Sodium thiosulfate at a concentration of 5 mM was then added to the previous solution at a volume ratio of 1:1 under continuous stirring. The reaction mixture was allowed to settle for 40 min, and turbidity was observed as an indicator of S-NP formation. 

### 2.3. Transmission Electron Microscopy (TEM) and X-ray Diffraction (XRD) Analyses

The size and morphology of the prepared NPs were characterized using a transmission electron microscope (JEOL, JEM 2100F, Tokyo, Japan) at 200 kV. A drop of the colloidal solution was used for specimen preparation for the TEM analysis. This drop was loaded onto a 400-mesh copper grid coated with amorphous carbon film. The solvent was allowed to evaporate at room temperature [23]. 

The crystalline nature of the synthesized nanoparticles was analyzed using an X-ray diffractometer (Bruker D8 ADVANCE, Karlsruhe, Germany) for the further confirmation of S-NP and Cu-NP formation. The X-ray tube target was Cu, with a voltage and current of 40 kV and 30 mA, respectively.

### 2.4. Evaluation of the Antifungal Activity

#### 2.4.1. Isolation and Purification

Naturally diseased fruit/root samples were collected from the region of Bilqas, Mansoura, Egypt, based on the symptoms found on cucumber fruits and roots related to *B. cinerea* and *S. sclerotiorum* pathogens. Isolation trials from naturally diseased cucumber plants were conducted. Three isolates of both pathogens *Botrytis cinerea* and *Sclerotinia sclerotiorum* were obtained from the diseased samples, but the most virulent isolate of each pathogen was selected to conduct this study. The fruit and root pieces were sterilized in 0.5% sodium hypochlorite solution for 1–2 min, then washed in sterile distilled water. The pieces were then dried on sterilized filter papers and placed on PDA medium supplemented with antibiotics (chloramphenicol; 5 mg/L) in Petri dishes. The plates were then incubated at 25 °C in the dark for 4–6 days. The growing fungi were then transferred to new PDA plates. The recovered fungi were purified using single-spore or hyphal-tip techniques, as appropriate. Pure cultures of the recovered fungi were maintained on carrot agar slants in test tubes incubated at 30 °C to allow them to grow before being stored in a refrigerator at 5 °C as stock cultures [24,25]. Fungi were identified with the help of authenticity books [26,27]. 

#### 2.4.2. Pathogenicity Test

Pathogenicity tests of two fungal isolates obtained from naturally diseased cucumber plants were conducted on cucumber fruits to fulfill Kokh’s postulates. Healthy cucumber fruits were inoculated with agar plug (5 mm diameter) taken from the edge of 7-day-old colonies with pure cultures of the isolated *B. cinerea* and *S. sclerotiorum*. They were allowed to produce disease symptoms, and then the pathogens re-isolated onto PDA medium. Fungal pathogens were identified using the morphological characteristics of mycelia, conidiophores, conidia, and sclerotia, as described by Saharan and Mehta [28].

#### 2.4.3. In Vitro Antifungal Activity

S-NPs and Cu-NPs were suspended in PDA medium at different concentrations ranging from 5 to 100 µg/mL and then poured into Petri dishes (90 mm in diameter). After PDA cooling and solidification, 5 mm diameter agar discs from 7-day-old cultures of *B. cinerea* and *S. sclerotiorum* were placed in the center of the PDA-NPs Petri dishes and incubated at room temperature (20 ± 2 °C). When the fungal growth covered the whole plate in the NP-free PDA control plates [29], the colony diameter was measured across all treatments to evaluate the antifungal activity of the prepared NPs compared to CuSO_4_ (4000 µg/mL) and micro sulfur (MS) at a concentration of 1000 µg/mL. Three replicates (plates) were used for each treatment and the experiment was repeated twice. The percent inhibition of radial growth of the pathogens was calculated according to the following equation:Growth inhibition (%) = [(R_1_ − R_2_)/R_1_] × 100 
where R_1_ = radial growth of the pathogen in the positive control, and R_2_ = radial growth of the pathogen on the treated plate [30].

#### 2.4.4. In Vivo Antifungal Activity

The antifungal activities of Cu-NPs and S-NPs were evaluated in vivo against *B. cinerea* and *S. sclerotiorum.* The fruits of cucumber (*Cucumis sativus* var. Hesham F1 hybrid) were inoculated with these two pathogenic fungi. Fresh cucumber**s** were purchased from the market in Mansoura, Egypt. Healthy uniform-sized cucumber fruits (wound- and rot-free) were washed with tap water and dipped into 2% sodium hypochlorite for 2 min. They were then rinsed in sterile distilled water and air-dried on sterile filter paper. The cucumbers were dipped into colloidal solutions of S-NPs (25 and 50 µg/mL) and Cu-NPs (50 and 100 µg/mL), as well as MS and CuSO_4_ at concentrations of 1000 and 4000 µg/mL, respectively. Cucumbers dipped into sterile distilled water for 10 min were used as negative controls. The cucumbers were air-dried for 2 h after all treatments. Mycelial disks (5 mm in diameter), taken from the edge of 3-day-old cultures of *B. cinerea* and *S. sclerotiorum* grown on PDA, were placed at the tip of each cucumber fruit. Inoculated cucumber fruits without chemical treatments were used as positive controls. Healthy cucumber fruits sprayed only with sterile distilled water were used as negative controls. Cucumbers were placed in plastic boxes containing a fixed-size piece of cotton saturated with sterile distilled water to maintain high relative humidity. The cucumbers were incubated at two temperature levels (10 and 20 °C) to study the effect of incubation temperature on fruit rot development by the tested fungi. Three replicates (five cucumbers each) were used for each treatment and the experiment was repeated twice. The percentage of disease incidence (DI) was calculated six days after inoculation according to the following formula:DI (%) = (number of infected cucumbers/total number of cucumbers assessed) × 100

Disease severity (DS) was determined as lesion length (cm) after 6 days [31].

### 2.5. Determination of TPC and TSS

Total phenolic content (TPC) and total soluble solids (TSS) were determined to evaluate the effect of the prepared NPs and other treatments on the shelf life of cucumber fruits. TPC was determined according to the method described by Deng et al. using the Folin–Ciocalteu reagent [32]. First, 1 mL of the cucumber methanolic extract was added to 150 μL of Folin–Ciocalteu reagent (10%) (*v/v*). After 4 min, 120 μL of Na_2_CO_3_ (7.5%) was added. The mixture was incubated for 45 min in a dark at room temperature (23 ± 2 °C). The absorbance was read at 760 nm, and gallic acid was used as a reference. TPC was expressed as mg gallic acid equivalent per gram of fresh weight (mg GAE/g FW).

To determine TSS, every whole cucumber fruit was ground to obtain a homogeneous sample. Approximately 40 g of each fruit was extracted and centrifuged at 3000 rpm for 5 min. The supernatant was collected to determine the TSS content using a digital refractometer (PR-101α, ATAGO Co. Ltd., Tokyo, Japan) [33].

### 2.6. Cytotoxicity Test

The cytotoxicity levels of the prepared S-NPs and Cu-NPs were assessed using the 3-(4,5-dimethylthiazol-2-yl)-2,5-diphenyltetrazolium bromide (MTT) dye assay for both WI-38 and Vero cells. WI-38 cells were diploid human cells composed of fibroblasts derived from the lung tissue of a 3-month-gestation female fetus, whereas Vero cells were isolated from kidney epithelial cells of an African green monkey [34]. The prepared NPs were plated in a six-well tissue culture plate containing the tested cells. Then, plates were incubated for 24 h at 37 °C. After decantation of the growth medium, the cell monolayer was washed with washing media. Physical signs of cytotoxicity were observed in these cells. The tissue was picked up, followed by the addition of 20 µL of MTT at a concentration of 5 mg/mL phosphate-buffered saline. The mixture was then incubated at 37 °C and 5% CO_2_ for 5 h. Formazan, as MTT metabolite, was resuspended in 200 µL dimethyl sulfoxide under shaking for 5 min and measured at 560 nm, while the background was read at 620 nm and then subtracted.

### 2.7. Statistical Analysis

The data were statistically analyzed using the Costat system for Windows, Version 6.311 (CoHort software, Monterey, CA, USA). All multiple comparisons were first subjected to analysis of variance (ANOVA) (Gomez and Gomez, 1984). Significant differences among treatment means were determined with Duncan’s new multiple range test at *p* = 0.05 (Duncan, 1955) [35,36,37]. The standard error of the average was obtained based on the following equation and was found to be less than + (−) 0.3:SEX=S/√n
where *S* = sample standard deviation, and *n* = number of observations of the sample.

## 3. Results and Discussion

### 3.1. NP Characterization

#### 3.1.1. TEM Analysis

The micrographs obtained from the TEM analysis of both Cu-NPs and S-NPs confirmed the formation of copper and sulfur particles at the nanoscale, as shown in Figure 1a,b. The prepared Cu-NPs were in the range of 2–12 nm and well distributed in the colloidal solution. The S-NPs ranged from 10 to 50 nm in size and tended to be close to each other. Additionally, they were spherical in shape and coated with PVP, which was added at a high concentration of 3 g PVP/100 mL colloidal solution as a stabilizer (Figure 1a,b). In addition, a turbidity was observed after the preparation of S-NPs as an indicator for the formation of S-NP colloidal solution [14]. Histogram bins of Cu-NPs were 2 nm wide and centered at 4, 6, 8, 10, and 12 nm. Cu-NPs with diameters in the range of 7–9 nm were considered together as particles with a diameter of 8 nm, as illustrated in Figure 1c. Furthermore, the histogram bins of the S-NPs were 10 nm wide, and centered at 15, 25, 35, and 45 nm, as shown in Figure 1d. The Cu-NPs exhibited a higher specific area than the S-NPs owing to their smaller average size. Most of the Cu-NPs were around 8 nm, while approximately 50% of the S-NPs were around 15 nm, as illustrated by the NPs histograms in Figure 1c,d.

#### 3.1.2. XRD Analysis

XRD analysis was conducted to confirm the formation of the S-NPs and Cu-NPs. The prepared NPs in their colloidal solution were highly crystalline, as shown in Figure 2. The diffraction peaks of the S-NPs corresponded to 113, 220, 222, 026, 206, 313, 044, and 317 crystal planes, while the Cu-NPs crystal planes were 111 and 220. The obtained diffraction peaks confirmed the formation of S-NPs and Cu-NPs based on their crystalline nature [38,39]. The XRD peak around 64° may be attributed to the 220 crystal plane of Cu_2_O, as a result of the Cu-NPs’ oxidation during the drying process before the XRD analysis [40,41], while sodium oxalate can be formed as a product of the reaction between oxalic acid and sodium thiosulphate during S-NP preparation. Therefore, diffraction peaks of sodium oxalate were observed and were attributed to 002, 102, 013, 111, 004, 014, 113, 121, 122 and 123 crystal planes [42]. *B. cinerea* and *S. sclerotiorum* are able to secrete oxalate into their surroundings. In several cases, the secretion of oxalate is required for pathogenesis. Therefore, oxalate did not significantly affect the fungal growth of the tested fungi [43,44].

### 3.2. Evaluation of the Antifungal Activity

#### 3.2.1. In Vitro Antifungal Activity

The fungicidal activities of the S-NPs and Cu-NPs were tested against both *B. cinerea* and *S. sclerotiorum* pathogens and are expressed as percent growth inhibition. The obtained results showed that S-NPs had the highest antifungal activity compared to other treatments, as listed in Table 1. However, the antifungal activity increased as the NP concentration increased. S-NPs at concentrations of 75 and 100 µg/mL led to a significant increase in the growth inhibition percentage, up to 94.12% in comparison with the positive control. On the other hand, a high concentration of MS (1000 µg/mL) was even less effective at inhibiting *B. cinerea* growth than the lowest concentration of S-NPs (5 µg/mL) and was equal in effect with 25 µg/mL S-NPs to inhibit *S. sclerotiorum*, indicating the effect of particle size on antifungal activity, which significantly increased as the size decreased. Similar results were obtained by Chantongsri et al., who studied the antifungal effects of microsulfur and S-NPs against nine different strains of fungi. They found that in all cases, the antifungal activity of S-NPs around 25 nm was 4–9 times higher than that of microsulfur, with an average particle size of 8 µm, confirming that the decrease in sulfur particle size maximizes antifungal activity [45]. Cu-NPs at low concentrations showed less antifungal activity than that of S-NPs, while both sulfur and copper nanoparticles at the high concentration of 100 µg/mL had an equal effect on pathogen growth inhibition. The green-synthesized Cu-NPs at a concentration of 100 µg/mL led to a significant inhibition of *B. cinerea* and *S. sclerotiorum* growth up to 94.12 and 92.48%, respectively, while CuSO_4_ inhibited their growth by only 58.82 and 77.44%, respectively, at 40 times higher concentration (4000 µg/mL), as shown in Table 1. Ouda studied the effect of Cu-NP concentration on the growth of *B. cinerea* and reported that the pathogen growth was minimized with the increase in Cu-NPs from 1 to 15 µg/mL which confirmed the obtained results in this study [46].

The obtained results indicate that the prepared NPs were more efficient than Topsin-M 70 WP, which is a common fungicide in the agricultural field. Topsin-M is a synthetic pesticide that includes thiophanate-methyl as an active component at a concentration of 70%. The repeated use of Topsin-M induces toxicity in various crop fields. Therefore, there is an urgent need to develop alternative eco-friendly antifungal agents such as S-NPs [47]. Based on the above, S-NPS and Cu-NPs showed outstanding antifungal activities against the tested pathogens and were more applicable than CuSO_4_, MS, and Topsin-M 70 WP, as illustrated in Table 1 and Figure 3. The high antifungal activity of NPs may be due to their plasmon resonance and high specific area, which increases as the size decreases [48].

#### 3.2.2. In Vivo Antifungal Activity

After a standard 3D infection test with *B. cinerea* and *S. sclerotiorum*, the disease incidence and disease severity developed on cucumber fruits by the tested fungi under all treatments were observed and evaluated at two temperature levels (10 and 20 °C), as shown in Table 2. Based on the results obtained from the in vitro fungicidal activity test, two concentrations of S-NPs (25 and 50 µg/mL) and Cu-NPs (50–100 µg/mL) were selected to test their effect on DI and DS caused by *B. cinerea* and *S. sclerotiorum* on cucumber fruits. 

With regard to the disease developed by *B. cinerea* on cucumber fruits at both temperature levels, there were no significant differences among all antifungal treatments at the tested concentrations, as listed in Table 2. However, it should be noted that copper sulfate was used at a concentration of 40–80 times that of nano-copper, and microsulfur was used at a concentration of 20–40 times that of nano-sulfur. However, DS developed on cucumber fruits treated with all antifungal treatments was significantly lower than that developed on the positive control at both incubation temperature levels, as illustrated in Table 2. 

With regard to the disease developed by *S. sclerotiorum* on cucumber fruits, the incubation temperature had substantial effect, as no disease developed at all at 10 °C when Cu-NPs, S-NPs, and CuSO_4_ were used, unlike the MS and positive control treatments. At 20 °C, all treatments had same DI (no significant differences among them) but all treatments significantly reduced DS compared to the positive control (Table 2). It can be concluded that, as the temperature decreased from 20 °C to 10 °C, the DS was notably minimized; this may indicate weak fungal resistance to various treatments at low temperature. Terefe et al. reported that the highest growth rate of *B. cinerea* was recorded at 22 °C, whereas Ahlem et al. reported that the highest growth of *B. cinerea* was observed at temperatures in the range of 15–25 °C [49,50], confirming the results listed in Table 2.

In general, DI may be high in a treatment while DS in the same treatment is trivial. For example, DI can 100% (which means that all fruits show some disease symptoms) while the DS is still very low (the area affected by the disease is small), as shown in Table 2 and Figure 4. The outstanding antifungal activity of S-NPs may be due to their high specific area resulting from the presence of sulfur particles at the nanoscale [48]. In addition, *B. cinerea* was more resistant to the different treatments used in this study and showed higher DI and DS on cucumber than *S. sclerotiorum*. 

All treatments, including S-NPs, Cu-NPs, CuSO_4_ and micro sulfur, minimized DS caused by *B. cinerea* or *S. sclerotiorum* at different temperatures in comparison with the positive controls, except for the microsulfur treatment, which did not show a significant change in the DS of *S. sclerotiorum* at 10 °C. Furthermore, a significant decrease in DI was observed only at 10 °C for NPs and CuSO_4_ treatments on the cucumber fruits inoculated with *S. sclerotiorum*. 

### 3.3. Total Phenolic (TPC) and Total Soluble Solids (TSS) Content

The TPC of the positive control significantly increased as a result of fungal infection compared to that of the negative control, as listed in Table 3. On the other hand, sulfur treatments led to a significant decrease in the TPC in comparison with the positive control, indicating the role of MS and S-NPs in reducing oxidative stress after fungal infection [51]. Although Cu-NPs minimized the DS, cucumber fruits treated with Cu-NPs at the concentration of 100 µg/mL, and inoculated with either *B. cinerea* or *S. sclerotiorum* showed the highest TPC compared to other treatments, as illustrated in Table 3. This significant increase in TPC with Cu-NPs treatment may be due to a reduction in the adverse effects caused by the tested fungi, which reduces the loss of TPC in cucumber fruits [52].

Similar studies have reported a significant increase in TPC as a result of inoculation with pathogens (fungi, bacteria, or viruses). The infection causes the upregulation of various phenolic components, leading to an increase in TPC [53,54]. 

The results also showed that the TSS was significantly reduced in cucumber fruits infected with *B. cinerea* or *S. sclerotiorum* in comparison with the negative control (healthy fruits) (Table 3). In addition, the infection with *B. cinerea* led to a greater decrease in TSS content (0.7%) than *S. sclerotiorum* (2.8%) because of the high DS caused by *B. cinerea* compared to that of *S. sclerotiorum*. Our findings are confirmed by many studies which report that TSS content decreases with an increase in DS and respiration rate [55,56]. The reduction in TSS content after fungal infection may be due to the harmful effects of the fungi, which accelerate the respiration rate of the infected fruits and lead to the consumption of the soluble solids found in those fruits [57].

Moreover, all treatments led to a significant increase in TSS content, compared with the positive control. Cucumber fruits treated with S-NPs exhibited the highest increase in TSS content compared to the other treatments used in this study, as shown in Table 3. 

### 3.4. Cytotoxicity of the Synthesized Nanoparticles

The cytotoxicity of the prepared nanoparticles was evaluated using the MTT assay on both WI-38 and Vero cell lines. MTT can be reduced to formazan by metabolically active cells via reducing the enzymes excreted from viable cells [58]. The cytotoxicity of the nanoparticles on WI-38 cells was higher than that on Vero cells, indicating that the harmful effect of the tested nanoparticles on the kidney was less than their effect on the lung. The obtained results showed that S-NPs exhibited low cytotoxicity against WI-38 and Vero cells compared to Cu-NPs. In addition, S-NPs at a concentration of 25 µg/mL exhibited less than 5% cytotoxicity, as illustrated in Table 4. Therefore, these results confirm the applicability of S-NPs as eco-friendly agents against fungal plant diseases.

It was noticed that cytotoxicity increased gradually with the increase in nanoparticle concentration, especially in the case of Cu-NPs, as shown in Table 4 and Figure 5. IC_50_ values of the prepared Cu-NPs were 10.79 and 3.89 µg/mL for WI-38 and Vero cells, respectively. The calculated IC_50_ values of the Cu-NPs illustrate that it is necessary to use protective measures when handling or spraying Cu-NPs on plants; this is not the case when using S-NPs. The generation of reactive oxygen species (ROS) is the most well-known mechanism of NP cytotoxicity. ROS induce oxidative stress, which leads to cells failing to maintain cellular physiological redox-regulated functions, resulting in DNA damage and variation in cell motility [59].

Cell shrinkage, nuclear condensation, and fragmentation are the morphological parameters that relate to cytotoxicity [60]. These changes were observed in the tested cells, and increased with the increasing concentration of nanoparticles, as shown in Figure 5. With the increase in Cu-NP concentration, the typical shape of WI-38 and Vero cells was lost, while cell density decreased. However, no notable changes were observed with S-NPs at concentrations below 25 µg/mL. The simple preparation method of S-NPs, as well as their chemical stability in aqueous dispersions and selective toxicity, enhance the applicability of S-NPs in many aspects, especially in the agricultural sector [61].

## 4. Conclusions

This study showed the potential of using NPs as antifungal agents to combat fungal plant diseases owing to their antifungal properties. S-NPs and Cu-NPs significantly decreased *B. cinerea* and *S. sclerotiorum* growth at relatively low concentrations (5–100 µg/mL) compared with MS, CuSO_4_, and Topsin-M, as common fungicides, at concentrations of 1000, 4000, and 1000 µg/mL, respectively. The formation of the prepared NPs was confirmed by TEM and XRD analyses. S-NPs and Cu-NPs were in the range of 10–50 nm, and 2–12.5 nm, respectively. They effectively suppressed fungal infections and increased the cucumber shelf life, as confirmed by TPC and TSS analyses. Sulfur treatments, either as MS or S-NPs, led to a significant decrease in TPC compared to the positive control, indicating that sulfur reduces oxidative stress. Fungal infection leads to a notable deterioration in cucumber fruits and shortens their shelf life. S-NPs were more effective against fungal infections and had lower cytotoxicity than Cu-NPs. In addition, they represent an eco-friendly and sustainable way to control fungal infections as a result of their low cytotoxicity.

## Figures and Tables

**Figure 1 jof-08-00412-f001:**
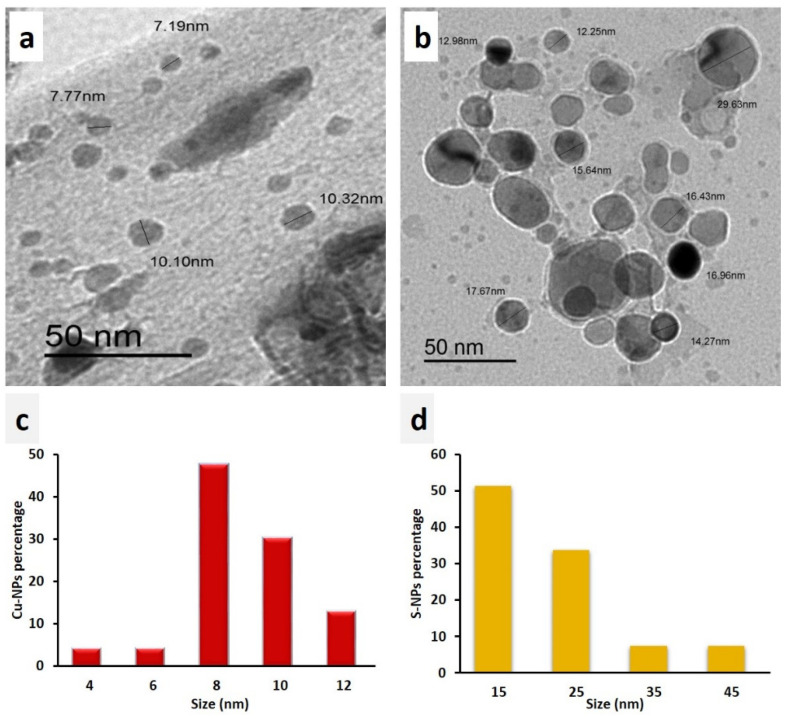
TEM micrographs of (**a**) Cu-NPs, (**b**) S-NPs, (**c**) Cu-NPs histogram, and (**d**) S-NPs histogram.

**Figure 2 jof-08-00412-f002:**
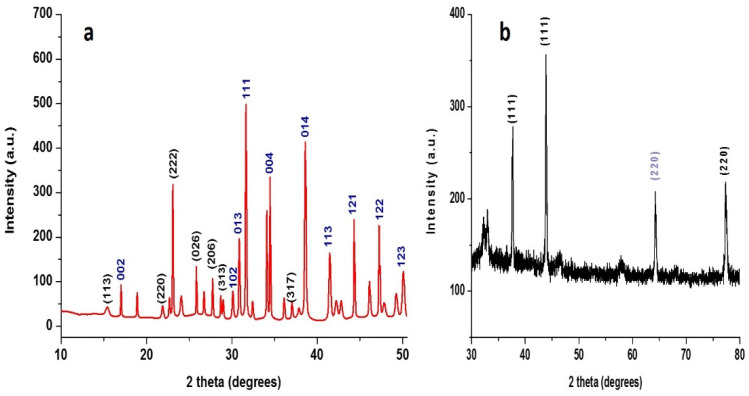
XRD patterns of prepared (**a**) S-NPs and (**b**) Cu-NPs.

**Figure 3 jof-08-00412-f003:**
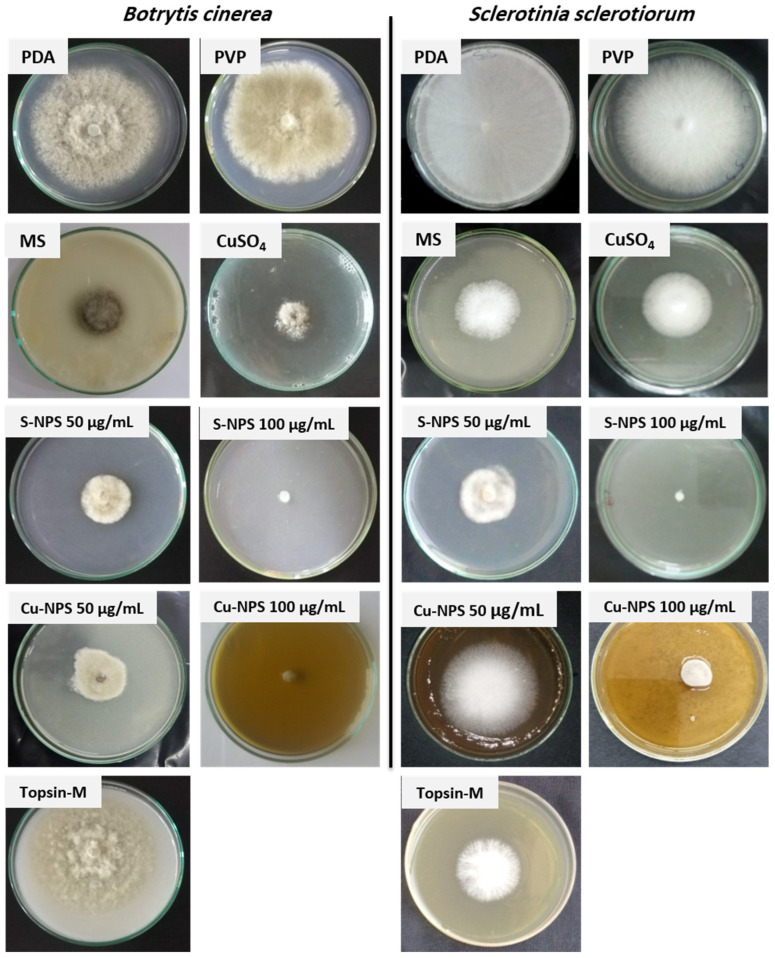
The antifungal activities of PVP (3 g/100 mL), MS (1000 µg/mL), CuSO_4_ (4000 µg/mL), Topsin-M (1000 µg/mL), S-NPs (50 and 100 µg/mL), and Cu-NPs (50 and 100 µg/mL) against *B. cinerea* and *S. sclerotiorum*.

**Figure 4 jof-08-00412-f004:**
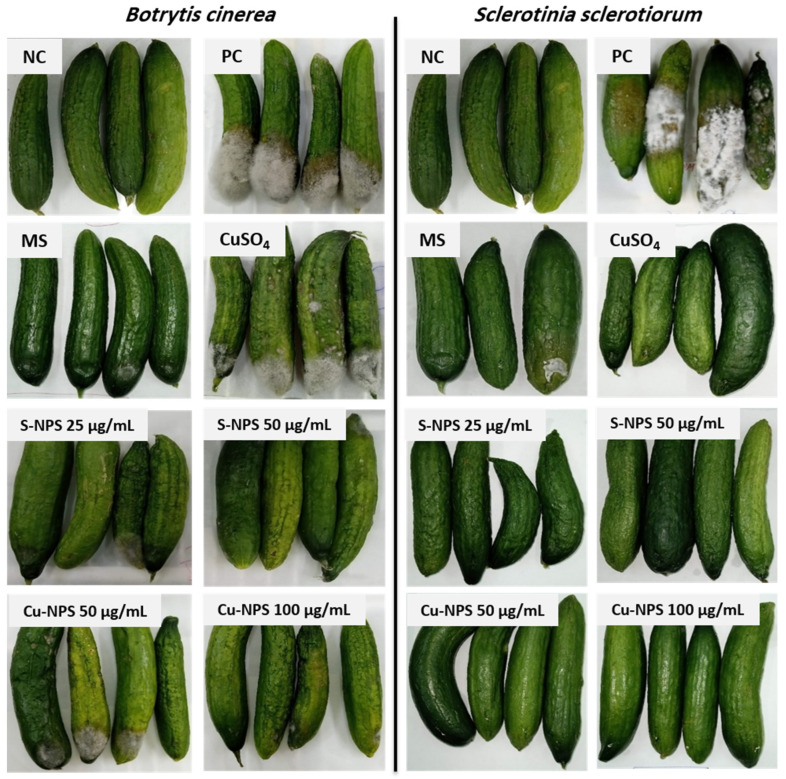
Effect of Cu-NPs and S-NPs at two concentrations (50 and 100 µg/mL) on disease incidence (DI) and disease severity (DS) caused by *B. cinerea* and *S. sclerotiorum* on cucumber fruits incubated at 20 °C in comparison with microsulfur (MS) (1000 µg/mL) and CuSO_4_ (4000 µg/mL). NC = negative control (cucumber fruits were dipped into sterile distilled water for 10 min); PC = positive control (cucumber fruits were inoculated with pathogens but without antifungal treatments).

**Figure 5 jof-08-00412-f005:**
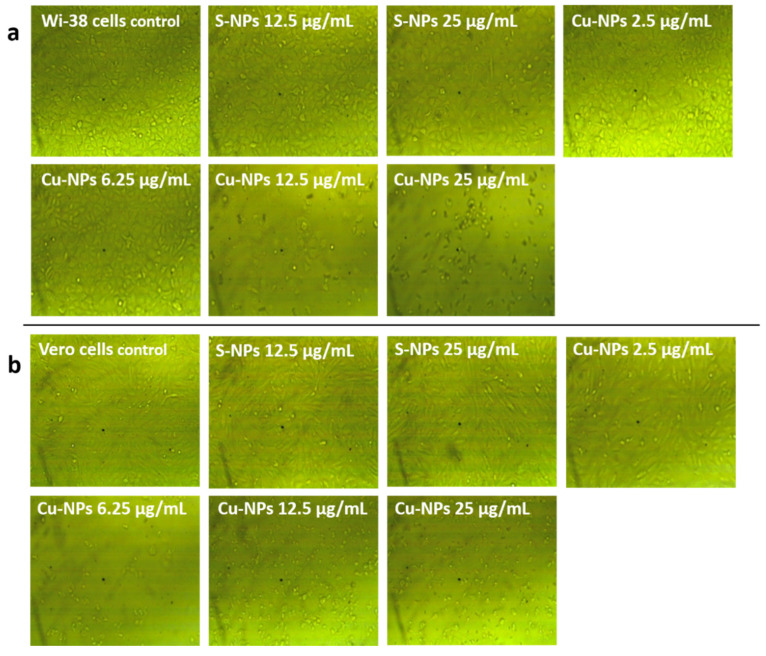
Effects of S-NPs and Cu-NPs at different concentrations on (**a**) WI-38 cells and (**b**) Vero cell viability.

**Table 1 jof-08-00412-t001:** Effect of Cu-NPs and S-NPs at different concentrations on *B. cinerea* and *S. sclerotiorum* growth on PDA.

Treatment	Concentration (µg/mL)	% Growth Inhibition	
		*Botrytis cinerea*	*Sclerotinia sclerotiorum*
S-NPs	5	76.47 f	54.89 k
	10	78.82 e	72.18 i
	25	80.00 d	83.08 e
	50	88.24 c	88.72 d
	75	90.98 b	90.60 c
	100	94.12 a	94.36 a
Cu-NPs	5	3.92 n	8.27 o
	10	13.73 k	13.53 n
	25	23.53 j	34.21 l
	50	47.06 i	63.91 j
	75	80.39 d	71.80 i
	100	94.12 a	92.48 b
CuSO_4_	4000	58.82 g	77.44 g
Micro sulfur (MS)	1000	58.04 h	82.33 f
Topsin-M 70 WP	1000	9.80 m	75.56 h
PVP (3 g/100 mL)		11.76 l	15.41 m

a–o Means with the same letter within the same column are not significantly different according to Duncan’s multiple range test (*p* = 0.05).

**Table 2 jof-08-00412-t002:** Effect of Cu-NPs and S-NPs at different concentrations on disease incidence (DI) and disease severity (DS) caused by *B. cinerea* and *S. sclerotiorum* on cucumber fruits incubated at two temperature levels (10 and 20 °C). ^a*^ DI was calculated six days after inoculation according to the following formula.

Treatment	Conc. (µg/mL)	*B. cinerea*				*S. sclerotiorum*			
		10 °C		20 °C		10 °C		20 °C	
		DI ^a*^ (%)	DS ^b*^ (cm)	DI (%)	DS (cm)	DI (%)	DS (cm)	DI (%)	DS (cm)
Cu-NPs	50	75 ab ^c*^	0.50 bc	75 ab	1 bcd	0 b	0 b	50 a	0.5 b
	100	25 ab	0.25 bc	50 ab	0.5 cd	0 b	0 b	50 a	0.5 b
S-NPs	25	75 ab	0.75 bc	75 ab	1.25 bc	0 b	0 b	50 a	0.5 b
	50	50 ab	0.50 bc	75 ab	0.75 cd	0 b	0 b	25 a	0.25 b
CuSO_4_	4000	100 a	1 b	100 a	2 b	0 b	0 b	50 a	1 b
Micro sulfur (MS)	1000	100 a	1 b	100 a	1.25 bc	25 ab	0.25 ab	50 a	0.5 b
Positive control ^c*^		100 a	1.75 a	100 a	4 a	100 a	1 a	75 a	3 a

^a*****^ DI (%) = (number of infected cucumber/total number of cucumber assessed) × 100. ^b*****^ DS was determined as lesion length (cm) after 6 days [31].^c*****^ Cucumber fruits were inoculated with pathogens only (without antifungal treatments). a–d Means with the same letter within the same column are not significantly different according to Duncan’s multiple range test (*p* = 0.05).

**Table 3 jof-08-00412-t003:** Effect of Cu-NPs and S-NPs at different concentrations on the total phenolic (TPC) and total soluble solids (TSS) contents in cucumber fruits inoculated with *B. cinerea* and *S. sclerotiorum.*

Treatment	Concentration (µg/mL)	*B. cinerea*		*S. sclerotiorum*	
		TPC (mg GAE/g FW)	TSS (%)	TPC (mg GAE/g FW)	TSS (%)
Cu-NPs	50	0.137 c	2.0 e	0.174 a	3.0 d
	100	0.152 a	3.8 b	0.169 a	3.4 c
S-NPs	25	0.064 f	3.9 a	0.109 d	3.3 c
	50	0.066 f	3.0 d	0.108 d	3.56 b
CuSO_4_	4000	0.141 b	3.2 c	0.093 d	3.0 d
Microsulfur	1000	0.104 e	3.0 d	0.108 d	3.4 c
Positive control ^a*^		0.143 b	0.7 f	0.146 b	2.8 e
Negative control ^b*^		0.129 d	4.0 a	0.129 c	4.0 a

^a*****^ Cucumber fruits were inoculated with pathogens only (without antifungal treatments).^b*****^ Healthy cucumber fruits were sprayed only with sterile distilled water. a–f Means with the same letter(s) within the same column are not significantly different according to Duncan’s multiple range test (*p* = 0.05).

**Table 4 jof-08-00412-t004:** Cytotoxicity levels of S-NPS and Cu-NPs on WI-38 and Vero cells.

Tested NPs	Concentration (µg/mL)	WI 38 Cells		Vero Cells	
		Viability (%)	Cytotoxicity (%)	Viability (%)	Cytotoxicity (%)
S-NPs	25	79.45 c	20.55 c	95.4 b	4.60 e
	12.5	98.48 a	1.52 e	99 a	1.00 f
	6.25	98.98 a	1.02 e	99.14 a	0.86 f
	2.5	99.77 a	0.23 e	99.62 a	0.38 f
	1.25	100 a	0.00 e	100 a	0.00 f
	0.625	100 a	0.00 e	100 a	0.00 f
Cu-NPs	25	6.81 e	93.19 a	11.24 f	88.76 a
	12.5	32.52 d	67.48 b	15.02 e	84.98 b
	6.25	91.55 b	8.45 d	16.02 d	83.98 c
	2.5	98.82 a	1.18 e	60.99 c	39.01 d
	1.25	99.09 a	0.91 e	99.2 a	0.80 f
	0.625	99.18 a	0.82 e	100 a	0.00 f

a–f Means with the same letter(s) within the same column are not significantly different according to Duncan’s multiple range test (*p* = 0.05).

## Data Availability

The data presented in this study are available on request from the corresponding author.

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
