# Peer review of "Antifungal Activities of Sulfur and Copper Nanoparticles against Cucumber Postharvest Diseases Caused by Botrytis cinerea and Sclerotinia sclerotiorum"

_jof, 2022, doi:10.3390/jof8040412_

Round 1

Reviewer 1 Report

Was the copper sulfate used to prepare the precursor hydrate or was it anhydrous?

Same question for sodium thiosulfate – hydrate or not? ... Please write its chemical formula in the text.

In particular, how have the methods for synthesizing S and Cu nanoparticles been improved compared to the methods cited?

Why was the concentration of copper precursor chosen 0.5 M, i.e. 500 mM and for sulfur 5 mM only? Is there a practical reason?

You are writing about monitoring turbidity in the case of S, but in the discussion I did not notice that the results were discussed ...

It would be better to put the TEM and XRD method in one chapter, moreover, when there is only one sentence for XRD.

Chapter title 2.3. is repeated twice.

In the methods, it would be useful to explain why TPC and TSS are determined.

How many nanoparticles were evaluated and in which SW to determine the size distribution?

I find a serious problem in the interpretation of diffraction patterns. The method does not specify the source (lamp). Some peaks (significant) for both types of nanoparticles are not identified at all in the pattern, so it cannot be said that the antifungal effect is caused / supported only by these zero-valent nanoparticles. Evaluating Muller indexes only is insufficient. There is no information regarding identified phases neither from PDF database nor relevant literature sources.

It is necessary to consult English, especially in the chapter - discussion of results - very often there is a decrease / increase, in one sentence, for example, 4 times.  Comments in parentheses should be included in the text.

In the part - introduction - I would expect a broader discussion about the real application of nanoparticles in the conditions of real cultivation (in-field) and risks - residues, accumulation in the environment and risk for consumers….

Author Response

Comment (1)

Was the copper sulfate used to prepare the precursor hydrate or was it anhydrous?

Reply

Copper (II) sulfate pentahydrate was used as a precursor for Cu-NPs, and was mentioned in the revised manuscript in the section of 2.1. with its chemical formula

Comment (2)

Same question for sodium thiosulfate – hydrate or not? ... Please write its chemical formula in the text.

Reply

Sodium thiosulfate pentahydrate was utilized as a precursor for S-NPs, and was reported also in the revised manuscript in the section of 2.1. with its chemical formula

Comment (3)

In particular, how have the methods for synthesizing S and Cu nanoparticles been improved compared to the methods cited?

Reply

The concentration of copper sulphate was changed to be 0.5 M instead of 0.2 M to increase the amounts of the prepared Cu-NPs to maximize the applicability of Cu-NPs, especially in the industrial scale in advance. In addition, the concentration of PVP was changed to be 6 g/100 mL to maintain the Cu particles at the nano scale after the increase in copper concentration.

In the case of S-NPs preparation, polyvinylpyrrolidone (PVP) was used instead of other surfactants mentioned in the cited paper due to the low cytotoxicity of PVP, which enhances the applicability of the prepared S-NPs as an eco-friendly antifungal agent, specifically at the post-harvest.

Comment (4)

Why was the concentration of copper precursor chosen 0.5 M, i.e. 500 mM and for sulfur 5 mM only? Is there a practical reason?

Reply

Several previous studies and the cited papers tested the effect of the precursor concentration on the obtained NPs size. It was found that the increase in Sodium thiosulphate concentration above 5 mM leads to a significant aggregation and form large particles not at nano scale as reported by   Chaudhuri et al.

“Chaudhuri, R.G.; Paria, S. Synthesis of sulfur nanoparticles in aqueous surfactant solutions. Journal of colloid and interface science 2010, 343, 439-446.”

Comment (5)

You are writing about monitoring turbidity in the case of S, but in the discussion I did not notice that the results were discussed ...

Reply

The observation of S-NPs turbidity was illustrated in the results and discussion section with a reference.

Comment (6)

It would be better to put the TEM and XRD method in one chapter, moreover, when there is only one sentence for XRD.

Reply

XRD method was put with TEM analysis in the same section.

Comment (7)

Chapter title 2.3. is repeated twice.

Reply

Thank you for your revision. We modified the title and removed the repeated title.

Comment (8)

In the methods, it would be useful to explain why TPC and TSS are determined.

Reply

The TPC and TSS were determined to evaluate the effect of the prepared NPs and other treatments on the shelf-life of the cucumber fruits. This explanation was added in the section 2.5.

Comment (9)

How many nanoparticles were evaluated and in which SW to determine the size distribution?

Reply

More than hundred particles of the prepared NPs obtained from various TEM micrographs were evaluated. Their diameters were determined using image software, as described by Sumadiyasa and Manuaba, and then the size distribution was calculated according to specific bins as described in the 3.1.1. section in the results and discussion.

Comment (10)

I find a serious problem in the interpretation of diffraction patterns. The method does not specify the source (lamp). Some peaks (significant) for both types of nanoparticles are not identified at all in the pattern, so it cannot be said that the antifungal effect is caused / supported only by these zero-valent nanoparticles.

 Evaluating Muller indexes only is insufficient. There is no information regarding identified phases neither from PDF database nor relevant literature sources.

Reply

The X-ray tube target was Cu with a voltage and current of 40 kV, and 30 mA, respectively. This explanation was also added in the section 2.3.

Regarding non-identified peaks, sodium oxalate is formed as a product as a result of the reaction between oxalic acid and sodium thiosulphate according the following equation:

H2C2O4 +Na2S2O3 → Na2C2O4 + S + SO2 + H2O

Therefore, XRD patterns of sodium oxalate appeared in the obtained XRD spectrum. The peaks were identified using high score analysis software, as shown in the following figure for further confirmation. XRD peak around 64⁰ may be attributed to Cu2O, as a result of Cu-NPs oxidation during drying process for XRD analysis. This explanation was added to the revised manuscript.

Comment (11)

It is necessary to consult English, especially in the chapter - discussion of results - very often there is a decrease / increase, in one sentence, for example, 4 times.  Comments in parentheses should be included in the text.

Reply

It is corrected

Comment (12)

In the part - introduction - I would expect a broader discussion about the real application of nanoparticles in the conditions of real cultivation (in-field) and risks - residues, accumulation in the environment and risk for consumers….

Reply

Introduction was modified as required.

Reviewer 2 Report

Review report of jof-1630082-peer-review-v1

Abstract:

More part of methodology and very less about results, the authors did not give conclusions of the study

Introduction:

Lack of consistency, the authors did not give any citation about the traditional management of both these postharvest pathogens of cucumber, strong justification on the use of nanoparticles is required in the introduction section

Line 37 reference is needed

Line47-48, add reference, how (cost is minimized) by the application of nanoparticles in place of fungicides?????

Line 60 (cost-effective, and adequate fungicides against B. cinerea and S. sclerotiorum on cucumbers), supportive literature?

What was the missing gap or why the authors selected these two nanoparticles??, any supportive literature?

Materials and Methods:

The detail of collection sites of disease samples is missing lines 103-104

Did the authors get one isolate per fungus from the disease samples??????

Line 110-111 reference missing

Line 117, is plug was placed to check the pathogenicity???

Line 139, variety/cultivar of cucumber????

Line 162 How much volume?????? (specific volume)

How the isolated fungi were identified, should be with the help of some taxonomic key, or published book???

Line 148, why to put the plug at the tip of each fruit????

Line 75 Liu et al., it is not according to the style of the journal, should be in numeric

The botanical name of cucumber (Cucumis sativus L.)

  1. cinerea can infect leaves, stem and fruit, how only the treatment of fruits with nanoparticles can be representative of disease management????

Why (10 and 20°C) were selected???

2.7. Data analysis

More elaborative statistically analysis is needed

Results

Table 2 needs to be explained more comprehensively as most treatments have non-significant impact on disease incidence (DI) and disease severity (DS) caused by B. cinerea and S. sclerotiorum

Discussion

The discussion section must be separated from the result section to make the paper more understandable.

On the bases of above, I strongly believe that this manuscript in present shape is not suitable to be published in JoF

Author Response

Author’s Responses to the Reviewers Comments

Journal

Journal of Fungi

Manuscript ID

Manuscript ID: jof-1630082

Title of Paper

Antifungal activities of sulfur and copper nanoparticles against cucumber postharvest diseases caused by Botrytis cinerea and Sclerotinia sclerotiorum

I am very much thankful to the reviewer (s) for their deep and thorough review. I have revised my present research paper in the light of their useful suggestions and comments and I have tried to address their comments and I hope my replies have addressed the answers of their comments to a level of their satisfaction. Each of their concerns has been addressed as outlined below and in separate file with modification made.

Response to Reviewer 2 comment

Comment (1)

Abstract: More part of methodology and very less about results, the authors did not give conclusions of the study

Reply

Abstract was modified and more conclusions were included.

Comment (2)

Lack of consistency, the authors did not give any citation about the traditional management of both these postharvest pathogens of cucumber, strong justification on the use of nanoparticles is required in the introduction section

Reply

Citation about the traditional management and NPs justification were included in the revised manuscript.

Comment (3)

Line 37 reference is needed

Reply

Reference was added

Comment (4)

Line47-48, add reference, how (cost is minimized) by the application of nanoparticles in place of fungicides?????

Reply

NPs exhibit high specific area. Therefore, the use of NPs minimizes the required amounts of antifungal agents. "This explanation was mentioned in the introduction chapter"

In addition, the obtained results confirmed the low cost of S-NPs comparing to microsulfur. S-NPs (100 µg/mL) inhibited about 94% of Botrytis cinerea growth, while microsulfur at higher concentration of 1000 µg/mL led to a lower inhibition (58%), as listed in Table 1. The use of NPs require small quantities; and therefore, they are considered cost-effective antifungal agents.

Comment (5)

Line 60 (cost-effective, and adequate fungicides against B. cinerea and S. sclerotiorum on cucumbers), supportive literature?

Reply

The authors illustrated that preparation of cost-effective and eco-friendly antifungal NPs is the aim of this study, which is supported by the obtained results as reported in the results and discussion chapter.

Comment (6)

What was the missing gap or why the authors selected these two nanoparticles??, any supportive literature?

Reply

Sulfur can be considered a highly efficient fungicide that is used in agriculture either as a fertilizer or fungicide. While, copper compounds are used widely as fungicides in agricultural field, as mentioned in the introduction chapter. Therefore, the evaluation of antifungal activities of sulphur and copper at nanoscale can attract the interest of scientific and industrial communities.

Comment (7)

The detail of collection sites of disease samples is missing lines 103-104

Reply

It is added in the revised manuscript.

Samples were collected from a farm in Bilqas region, Dakahlia, Egypt

Comment (8)

Did the authors get one isolate per fungus from the disease samples??????

Reply

Three isolates of both pathogens Botrytis cinerea and Sclerotinia sclerotiorum were obtained from diseased samples, but the most virulent isolate of each pathogen was selected to conduct this study.

This explanation was also mentioned in section 2.4.1.

Comment (9)

Line 110-111 reference missing

Reply

The reference was added in the revised manuscript.

Comment (10)

Line 117, is plug was placed to check the pathogenicity???

Reply

Yes, agar plugs (5 mm diameter) taken from the edge of 7–day-old colonies of pure cultures of the isolated B. cinerea and S. sclerotiorum were used. This was explained in the text in the revised manuscript using "Track Changes" tool.

Comment (11)

Line 139, variety/cultivar of cucumber????

Reply

It is mentioned in sction 2.4.4. in the revised manuscript.

Comment (12)

Line 162 How much volume?????? (specific volume)

Reply

Methanolic extract volume was mentioned in section 2.5. in the revised manuscript.

Comment (13)

How the isolated fungi were identified, should be with the help of some taxonomic key, or published book???

Reply

Fungi were identified with the help of the following books, as reported into the text of the revised manuscript in section 2.4.1.:

Barnett, H. L. and Hunter, B. B. (1999). Illustrated Genera of Imperfect fungi. 4th edition. St. Paul, Minnesota, APS Press. The American Phytopathological Society, 76 pp.

Hanlin, R.T. (1999). Illustrated Genera of Ascomycetes. Vol. I, the American Phytopathological Society, 64, 65 pp.

Comment (14)

Line 148, why to put the plug at the tip of each fruit????

Reply

This is a standard method. Place of the plugs on fruits must be fixed and uniform among all treatments. 

Comment (15)

Line 75 Liu et al., it is not according to the style of the journal, should be in numeric

Reply

It is corrected

Comment (16)

The botanical name of cucumber (Cucumis sativus L.)

Reply

The botanical name was mentioned in the abstract in the revised manuscript.

Comment (17)

cinerea can infect leaves, stem and fruit, how only the treatment of fruits with nanoparticles can be representative of disease management????

Reply

The aim of this study is to evaluate the antifungal activities of S-NPs and Cu-NPs against cucumber postharvest diseases. Therefore, the fungal growth was determined on the fruits.

Comment (18)

Why (10 and 20°C) were selected???

Reply

We used different temperatures to study the effect of temperature on the fungal growth and the antifungal activities of the prepared NPs, as mentioned in section 2.4.4.

Comment (19)

More elaborative statistically analysis is needed

Reply

The standard error was also determined, as mentioned in section 2.7. in the revised manuscript.

Comment (20)

Table 2 needs to be explained more comprehensively as most treatments have non-significant impact on disease incidence (DI) and disease severity (DS) caused by B. cinerea and S. sclerotiorum.

Reply

An explanation was written in section 3.2.2. to illustrate the significant variations in DS and DI percentage values listed in Table 2.

Comment (21)

The discussion section must be separated from the result section to make the paper more understandable. On the bases of above, I strongly believe that this manuscript in present shape is not suitable to be published in JoF

Reply

Thank you so much for your valuable revision. Results may make little sense to most readers without interpretation. Therefore, we combined results and discussion sections.

Round 2

Reviewer 1 Report

I would like to thank the authors for their clarifications and data completion. recommend publishing the modified version. After their editing, the manuscript is suitable for publishing.